# Thriving under Salinity: Growth, Ecophysiology and Proteomic Insights into the Tolerance Mechanisms of Obligate Halophyte *Suaeda fruticosa*

**DOI:** 10.3390/plants13111529

**Published:** 2024-05-31

**Authors:** Bilquees Gul, Abdul Hameed, Muhammad Zaheer Ahmed, Tabassum Hussain, Sarwat Ghulam Rasool, Brent L. Nielsen

**Affiliations:** 1Dr. Muhammad Ajmal Khan Institute of Sustainable Halophyte Utilization, University of Karachi, Karachi 75270, Pakistan; ahameed@uok.edu.pk (A.H.); mzahmed@uok.edu.pk (M.Z.A.); thussain@uok.edu.pk (T.H.); sarwat99@live.com (S.G.R.); 2Department of Microbiology & Molecular Biology, Brigham Young University, Provo, UT 84602, USA; brentnielsen@byu.edu

**Keywords:** halophyte, ion homeostasis, oxidative stress, photosynthesis, proteomics, water relations

## Abstract

Studies on obligate halophytes combining eco-physiological techniques and proteomic analysis are crucial for understanding salinity tolerance mechanisms but are limited. We thus examined growth, water relations, ion homeostasis, photosynthesis, oxidative stress mitigation and proteomic responses of an obligate halophyte *Suaeda fruticosa* to increasing salinity under semi-hydroponic culture. Most biomass parameters increased under moderate (300 mmol L^−1^ of NaCl) salinity, while high (900 mmol L^−1^ of NaCl) salinity caused some reduction in biomass parameters. Under moderate salinity, plants showed effective osmotic adjustment with concomitant accumulation of Na^+^ in both roots and leaves. Accumulation of Na^+^ did not accompany nutrient deficiency, damage to photosynthetic machinery and oxidative damage in plants treated with 300 mmol L^−1^ of NaCl. Under high salinity, plants showed further decline in sap osmotic potential with higher Na^+^ accumulation that did not coincide with a decline in relative water content, *Fv*/*Fm*, and oxidative damage markers (H_2_O_2_ and MDA). There were 22, 54 and 7 proteins in optimal salinity and 29, 46 and 8 proteins in high salinity treatment that were up-regulated, down-regulated or exhibited no change, respectively, as compared to control plants. These data indicate that biomass reduction in *S. fruticosa* at high salinity might result primarily from increased energetic cost rather than ionic toxicity.

## 1. Introduction

The world population that currently stands at approximately 8 billion people is growing rapidly and is estimated to surpass the 9 billion mark by 2050 [1,2]. To feed the rapidly growing human population, an average annual increase of 44 million tons of food production will be required [3,4]. In contrast, reliance on irrigation for crop production alongside global climate changes are resulting in salinization of agricultural lands, which poses a major threat to our food security in the future [2,5,6]. According to some estimates, up to 50% of irrigated lands are already salinity-affected to some extent [7], and a further ~1.5 Mha of irrigated lands is degraded annually due to salinization [8,9]. Hence, in order to achieve food security in the future, there is an urgent need to employ multidisciplinary approaches including reclamation of saline soils, developing salinity-tolerant crops and utilization of naturally salinity-tolerant halophytes as non-conventional crops. Meanwhile, reliable solutions to the first two approaches are not available. Utilization of halophytes for economic purposes thus appears to be an immediate solution, which is both economically and ecologically feasible [10]. Hence, a thorough understanding of salinity tolerance mechanisms of halophytes is not only important to improve their crop potential but will also contribute in enhancing salinity tolerance of conventional crops [2,11,12].

Salinity primarily affects plants initially by imposing osmotic constraint due to reduction in the rhizosphere water potential, which is then accompanied by an ionic effect resulting from entry of excess Na^+^ and Cl^−^ into cells [13,14,15]. Most plants undergo cell plasmolysis (water loss) and disruption of metabolism (toxicity) due to osmotic and ionic constraints of salinity [13]. However, halophytes have evolved diverse mechanisms to deal with the aforementioned effects of salinity [16,17,18]. Compartmentalization of toxic ions in vacuoles and apoplasts with concomitant accumulation of organic compatible solutes such as quaternary ammonium compounds and sugars in the cytoplasm are considered important features of halophytes, which not only help in osmotic adjustment but also reduce ionic toxicity [16].

Salinity often results in biomass reduction, which is ascribed to a decline in CO_2_ assimilation in photosynthesis [19,20,21,22,23,24]. This decrease in net CO_2_ assimilation may result from stomatal closure to reduce transpiration and increase water use efficiency under salinity [25,26]. In many cases, prolonged exposure of plants to salinity may also affect biochemical reactions of CO_2_ assimilation, such as rubisco carboxylase/oxygenase activity, and regeneration of RuBP and triose phosphates [22,27]. These stomatal and/or biochemical limitations inhibit consumption of ATP and NADPH that results in over-reduction of photosynthetic machinery [28,29]. Excess excitation energy may damage the photosynthetic apparatus, namely D_1_ and D_2_ proteins of photosystem II [29,30]. In addition, leakage of energy and electrons from over-reduced photosynthetic machinery can also activate molecular oxygen, leading to the formation of reactive oxygen species (ROS), which, if not tightly regulated to low levels, can damage membranes and other important components of plant cells [23]. The ability to protect photosynthetic machinery and minimize ROS production under salt stress is another remarkable feature of halophytes [31,32]. Therefore, besides dealing with primary effects of salinity, salt tolerance could be related to the plant’s ability to maintain adequate photosynthetic gas exchange [33], protect photosynthetic machinery [32,34] and minimize oxidative damage resulting from excess ROS [35] under saline conditions.

*Suaeda fruticosa* is a leaf succulent perennial halophyte from the Amaranthaceae family, which is native to arid and semi-arid salt flats, salt marshes, and other stressful habitats in Pakistan and adjacent countries of the Saharo-Sindian and southern Irano-Turanian regions [36,37]. It is a shrub with variable leaf morphology differing in shape, size, degree of succulence and color ranging from dark green to purple (Flora of Pakistan—http://www.efloras.org/florataxon.aspx?flora_id=5&taxon_id=242100197; accessed on 27 May 2024). There are many potential economic uses of this halophyte such as edible oil from seeds [38], ‘sajji’ (domestic soap) from burnt leaves [39], biodiesel production [40], forage for camels [41], bioremediation/reclamation of salinity [42] or toxic metal reduction [43,44] in affected lands. Different vegetative parts of *S. fruticosa* also have antibacterial, antiophthalmic, hypolipidemic, anticancer and hypoglycemic effects [37,45,46,47,48]. It is an obligate halophyte with survival in up to 1000 mmol L^−1^ of NaCl (~1.5-fold seawater equivalent) salinity and growth stimulation in 300 mmol L^−1^ of NaCl (~0.5-fold seawater equivalent) by efficient osmotic adjustment, ROS balancing and Na^+^ homeostasis [49,50,51]. A recent paper reported CO_2_ uptake and chlorophyll a fluorescence of *S. fruticosa* grown under diurnal rhythm and after transfer to continuous dark conditions [52]. Another paper reported ‘C4 photosynthesis pathway enzyme (PPDK)’ response to a rise in CO_2_ [53]. However, effects of increasing salinity on photosynthetic gas exchange and chlorophyll a fluorescence of *S. fruticosa* are yet to be studied. In addition, information about changes in the protein profile of *S. fruticosa* under optimal and sub-/supra-optimal salinity levels is lacking. This study thus attempts to fill the aforementioned knowledge gaps by combining classical eco-physiological analyses with proteomics. The aim of this study is to provide comprehensive information about growth, water relations, ion homeostasis, photosynthesis, oxidative stress mitigation and proteomic responses of *S. fruticosa* under increasing salinity. More specifically, this study will provide baseline information about optimal performance of obligate halophyte *S. fruticosa* under moderate (300 mmol L^−1^ of NaCl) salinity and survival mechanisms under high (900 mmol L^−1^ of NaCl) salinity.

## 2. Results

### 2.1. Plant Growth

Biomass (dry and fresh weight of leaf, stem and root) of *S. fruticosa* was improved at moderate (300 mmol L^−1^ of NaCl) salinity (Table 1; Figure 1). High (900 mmol L^−1^ of NaCl) salinity significantly (*p* < 0.05) inhibited stem dry weight and total plant biomass as compared to the plants growing in moderate salinity and control treatments. However, the dry weight of leaves and root length were unaffected under all treatments (Table 1).

### 2.2. Water Relations

Leaf osmotic potential (ψ_S_) and xylem pressure potential (XPP) decreased (i.e., became more negative) with an increase in substrate salinity. However, leaf relative water content (RWC) was maintained under saline treatments (Table 1).

### 2.3. Cation Content and K^+^-Selectivity

As the salinity of the substrate increased, Na^+^ ion content spiked in both leaf and root tissues; the rise in leaf tissues was approximately five times greater than the increase in root tissues (Figure 2). With the exception of a drop in the Mg^2+^ ion content in leaves, the K^+^, Ca^2+^ and Mg^2+^ contents of leaves remained steady across salinity levels. On the other hand, K^+^ and Ca^2+^ contents in root tissues rose in response to salt treatments, peaking in roots treated with moderate salinity (Figure 2). The Na^+^/K^+^ ratios increased in leaves under moderate and high salinity; however, in root tissues increased values were found in higher salinity only (Figure 2). K^+^ selective absorption over Na^+^ was higher in 900 mmol L^−1^ of NaCl than in the non-saline control, but it remained below 300 mmol L^−1^ of NaCl. In contrast, selective transport of K^+^ towards shoots was higher in plants of 900 mmol L^−1^ of NaCl than 300 mmol L^−1^ of NaCl but similar to the non-saline control (Figure 2).

### 2.4. Photosynthetic CO_2_/H_2_O Gas Exchange, Chlorophyll Fluorescence and Pigments

A slight increase in net photosynthetic rate (*A*) was observed in moderate salinity in comparison to control treatment plants. However, *A*, stomatal conductance (*gs*) and intercellular CO_2_ (*Ci*) declined at higher salinity (Table 2). The net transpiration rate (*E*) increased in moderate salinity but was similar in control and high salt treatments. Water use efficiency (*WUE*) was unchanged across salinity, and dark respiration (*Rd*) increased in high salinity only (Table 2). The saturation point (*Is*) decreased with an increase in salinity; however, carbon compensation point (*Ic*) was the highest in higher salinity and the lowest in moderate salinity (Table 2).

The *Fv*/*Fm*, *NPQ* and *ETR* remained unaltered across salinity treatments. *Y*(*II*) declined in moderate salinity; however, *qP* decreased in both moderate and high salinities as compared to control plants (Figure 3; Table 2). Photosynthetic pigments (chl a, chl b, total chl and carotenoids) declined under salt treatments when compared to the control treatment. However, there was no difference for those pigments between moderate and high salinity. The ratios of chl a/b and chl/carotenoids remained unchanged in all treatments (Table 2).

### 2.5. Oxidative Damage Markers, Enzymatic and Non-Enzymatic Antioxidants

The content of H_2_O_2_ decreased in moderate salinity and was comparable to the non-saline control in high salinity treated plants (Figure 4). MDA content of the leaves decreased under saline conditions (Figure 4). Activities of SOD, CAT and GPX decreased, whereas those of APX and GR remained unaltered under moderate salinity. While activities of SOD, APX and GPX increased, those of CAT and GR remained unaffected under high salinity (Figure 4). The level of AsA did not vary across salinity treatments, while GSH content decreased only transiently in moderate salinity compared to control and high salinity treatment (Figure 4).

### 2.6. Differentially Expressed Proteins (DEPs) under Salt Treatments

Compared to the control plants, 83 DEPs were identified in the leaves of the moderate and high salinity treatments. DEPs are presented in the Appendix A. These DEPs were classified into 11 functional categories and identified as amino acid synthesis, carbohydrate metabolism, cofactor/coenzyme synthesis, defense, detoxifying and antioxidant, DNA-modifying, energy production and conversion, photosynthesis, post-translational modification and signal transduction, transcription and translation, and transport and structural proteins. In comparison to control plants, there were 22, 54 and 7 proteins in the optimum treatment, and 29, 46 and 8 proteins in the high salinity treatment that were up-regulated, down-regulated and unchanged, respectively. The highest number of DEPs were recorded in the functional category of photosynthesis (19 DEPs), followed by 18 DEPs in carbohydrate metabolism (Figure 5). However, DEPs were also recorded in the following functional categories; amino acid synthesis (7), cofactor/coenzyme synthesis (4), defense (1), detoxifying and antioxidant (4), DNA-modifying (3), energy production and conversion (8), post-translational modification and signal transduction (9), transcription and translation (7), and transport and structural (3).

## 3. Materials and Methods

### 3.1. Plant Growth and Culture

Seed-containing inflorescence twigs of *S. fruticosa* were collected from a large population growing in the coastal area of Hawkes Bay, Karachi, Pakistan (GPS coordinates: 24.52313 N, 66.52200 E). A plant specimen of the test species was submitted to the Herbarium of the Institute at University of Karachi with specimen voucher number DMAK-2020-01SF. Seeds were scrub-cleaned before sowing in garden soil to allow germination, and then seedlings were transplanted into small plastic mugs (7 cm diameter × 9 cm depth) under the green net house and irrigated with ½ strength Hoagland’s nutrient solution for 4 weeks [54]. Temperature range and average humidity during plant culture were 21.7 to 32.1 °C and 51.3%, respectively. Equal-sized seedlings (3–4 leaf) were transplanted into pots (13 cm diameter × 30 cm depth) filled with quartz sand and irrigated using the semi-hydroponic quick check system [55] for 2 weeks to establish the seedlings. NaCl was then added gradually with doses of 150 mmol L^−1^ in steps of 24 h intervals to avoid osmotic shock, in such a manner that the final concentrations of 300 and 900 mmol L^−1^ of NaCl were achieved on the same day. Irrigation solutions were changed every week to avoid nutrient deficiency; however, the evaporated water loss was balanced daily to avoid any change in NaCl concentration. Plants were harvested after four weeks of salinity treatment.

### 3.2. Biomass Measurements

Plant fresh weight (FW) was measured immediately following harvest. Dry biomass (DW) was measured after the samples were dried in an oven at 65 °C for 48 h, or until a consistent weight was obtained. Plant height, number of leaves, and leaf area were also measured.

### 3.3. Water Relations

Relative water content (RWC) of the leaves was estimated as RWC = (FW − DW)/(TW − DW) × 100, where TW is the turgid weight that was measured after placing leaf samples in distilled water overnight at 4 °C. Xylem pressure potential (XPP) was measured on twig cuttings near leaves in a Plant Water Status Console (Model 1000 PMS Instrument Co., Albany, NY, USA). Leaf osmotic potential (Ψ_s_) was measured on leaf sap extracts by using a vapor pressure osmometer (VAPRO 5520, Wescor Inc., Logan, UT, USA). In addition, the Ψ_s_ values of irrigated solutions were also measured.

### 3.4. Cation Content and Ion Selectivity

Leaves were frozen at −80 °C immediately after harvest. Later, samples were thawed and hand-squeezed to collect all the sap [56]. The sap was used to measure Na^+^, K^+^, Mg^2+^ and Ca^2+^ concentrations (mmol L^−1^ tissue sap) using an atomic absorption spectrometer (AA-700; Perkin Elmer, Valencia, CA, USA). The selective absorption (SA) and selective transport (ST) of K^+^, Mg^2+^ and Ca^2+^ over Na^+^ were calculated using the formulae of Pitman (1965) [7]:SA[X/Na^+^] = (root X/root Na^+^)/(soil X/soil Na^+^)
ST[X/Na^+^] = (shoot X/shoot Na^+^)/(root X/root Na^+^)
where ‘X’ represents the individual nutrient (K^+^, Mg^2+^ and Ca^2+^).

### 3.5. Photosynthetic Gas Exchange, Chlorophyll Fluorescence and Pigments

Photosynthetic CO_2_/H_2_O gas exchange was measured using a Li-COR 6400XT (LI-COR, Lincoln, NE, USA) instrument at 400 µmol m^−2^ s^−1^ of CO_2_, 33–36 °C and 45–60% relative humidity. A light response curve was constructed from 0 to 2000 µmol photon m^−2^ s^−1^ of photosynthetically active radiation (PAR). The net photosynthetic rate (*A*), stomatal conductance (*gs*), transpiration (*E*), intercellular CO_2_ (*Ci*) and water use efficiency (*A/E*) were measured on fully emerged leaves at saturated irradiation of each treatment under the constant light source. The dark respiration rate (*Rd*), compensation point (*Ic*) and saturation point (*Is*) were calculated according to [57].

A pulse-modulated chlorophyll fluorescence meter (PAM 2500, Walz, Effeltrich, Germany) was used to record chlorophyll fluorescence of healthy leaves. Leaves were dark-adapted for 30–40 min for measuring the minimal fluorescence (*Fo*) value after applying modulated light (<0.1 mol photon m^−2^ s^−1^), while the maximal fluorescence (*Fm*) value was recorded by imposing a saturating pulse of 10,000 mol photons m^−2^ s^−1^ for 0.8 s. *Fo* and *Fm* values were used to calculate the maximum photochemical quantum yield of PSII [*Fv*/*Fm* = (*Fm* − *Fo*)/*Fm*] [58]. Subsequently, leaves were continuously illuminated with actinic light to construct a rapid light curve (*RLC*). Steady-state (*Fs*) and maximal fluorescence (*Fm*′) were measured in light-adapted (1298, 900, 632, 300, 185, 107, 41 and 0) leaves. The minimal fluorescence level in light-adapted leaves (*Fo*′) was estimated following the method of [59]. The effective photochemical quantum yield of *PSII *(*YII*) was calculated as [*YII* = (*Fm*′ − *Fs*)/*Fm*′] [60]. Non-photochemical quenching of fluorescence (*NPQ*), which is proportional to the rate of constant heat dissipation, was calculated as *NPQ* = *Fm*/*Fm*′ − 1. The coefficient of photochemical quenching (*qP*) was calculated as (*Fm*′ − *Fs*)/(*Fm*′ − *Fo*′) [61,62]. *YII* was used for calculation of the linear electron transport rate (*ETR*; [63]) as *ETR* = *YII* × *PPFD* × 0.5 × 0.84, where *PPFD* is the Photosynthetic Photon Flux Density incident on the leaf; 0.5 is the factor that adopts equal distribution of energy between the two photosystems; and 0.84 is assumed leaf absorbance.

Fresh leaves were frozen in liquid nitrogen (0.5 g) and homogenized in 80% acetone using a chilled mortar and pestle. The homogenate was centrifuged at 3000× *g*, and the supernatant was collected to measure absorbance at 663.2, 646.8 and 470 nm (Beckman-Coulter DU-730, UV–vis spectrophotometer, Brea, CA, USA). The contents of chlorophyll a, b, total chlorophyll and total carotenoid were calculated with equations suggested by [64].

### 3.6. Enzymatic and Non-Enzymatic Antioxidants

Crude extracts of fresh leaves (0.5 g) were prepared according to Polle et al. (1994) [65]. Briefly, leaves were ground to a fine powder in liquid nitrogen and then homogenized in 50 mM of potassium phosphate buffer (pH 7.0) containing 5 mM of disodium ethylenediaminetetraacetic acid, 2% (*w*/*v*) polyvinyl-pyrrolidone and 1 mM of L-ascorbic acid. The homogenate was centrifuged at 12,000× *g* for 20 min at 4 °C, and the supernatant was used for assays of different antioxidant enzymes. Protein concentrations in the enzyme extracts were determined by the Bradford method [66] with bovine serum albumin (BSA) as standard. CAT activity was estimated according to the method of [67]. A decrease in H_2_O_2_ within a reaction mixture of 3 mL during 1 min at 240 nm (ε = 39.9 mM^−1^ cm^−1^) was measured. Ascorbate peroxidase (APX; EC 1.11.1.11) activity was assayed in a reaction mixture consisting of 50 mM of potassium phosphate buffer (pH 7.0), 0.2 mM of EDTA, 0.5 mM of ascorbate (ε = 2.8 mM^−1^ cm^−1^), 2 mM of H_2_O_2_ and 50 µL of enzyme extract in a final volume of 3 mL. The absorbance was measured at 290 nm [68]. Guaiacol peroxidase (GPX, EC 1.11.1.7) activity was assayed according to the method of Zaharieva [69] by measuring brown-colored tetraguaiacol formation at 470 nm (ε = 26.6 mM cm^−1^). Glutathione reductase (GR; EC 1.6.4.2) activity was determined according to the method of Foyer and Halliwell [70] in a reaction mixture containing 100 mM of Tris-HCl (pH = 7.8), 0.5 mM of GSSG, 0.03 mM of NADPH (ε = 6.2 mM cm^−1^), 5 mM of EDTA and 100 mL of enzyme extract, and the decrease in absorbance due to oxidation of NADPH in the presence of oxidized glutathione (GSSG) at 340 nm was recorded. The SOD activity assay was performed as described in [71]. The difference of absorbance at 560 nm under light was measured against the darkened control. The absorbance recorded in the absence of enzyme extract was taken as 100%, and the enzyme activity was calculated as the percentage inhibition per min. All enzyme activities are expressed as unit per mg protein (U mg^−1^ protein). A TCA extract was used to estimate ascorbic acid (AsA) and reduced glutathione (GSH). The supernatant was used to quantify GSH according to Guri [72]. Ascorbic acid (AsA) was determined by the method of Luwe et al. [73].

### 3.7. Oxidative Damage Markers

Fresh leaves (0.5 g) were ground in liquid nitrogen with a mortar and pestle and homogenized in 1% (*w*/*v*) trichloroacetic acid (TCA). Homogenates were centrifuged at 12,000× *g* for 20 min at 4 °C, and supernatant was separated to measure hydrogen peroxide (H_2_O_2_) and malondialdehyde (MDA) contents according to the methods of Loreto and Velikova, [74] and Heath and Packer [75], respectively.

### 3.8. Proteomic Analysis

Protein was extracted using a DTT-TCA-acetone precipitation method. Leaves were immediately frozen in liquid nitrogen after excision from the plant. One gram of ground leaf sample was homogenized with 2 mL 10% (*w*/*v*) of TCA in acetone containing 50 mM of dithiothreitol (DTT) and centrifuged at 14,000× *g* for 15 min at 4 °C. The precipitated proteins were pelleted and washed with ice-cold acetone containing 50 mM of DTT and 2 mM of EDTA. Washing was repeated until the supernatant was colorless. The protein pellet obtained was vacuum-dried, re-suspended in lysis buffer (8 M of urea, 2 M of thiourea, 4% CHAPS, 0.5% IPG buffer, 30 mM of DTT, 5 mM of PefaBloc) and vortexed at 33 °C for 2 h. Cell debris was removed by centrifugation at 14,000× *g* (4 °C, 30 min), and the supernatant was used for protein determination. Protein concentration was determined according to Bradford (1976), using a commercial dye reagent (Bio-Rad, Hercules, CA, USA) with bovine serum albumin as a standard.

Proteins were then subjected to MudPit (multidimensional protein identification technology) analysis using a Thermo LTQ Orbitrap XL ETD Hybrid Ion Trap-Orbitrap Mass Spectrometer (Thermo Fisher Scientific, Waltham, MA, USA) with an Eksigent nanoLC system and cooled autosampler for quantitative proteomics experiments. The results were analyzed using Mascot (Matrix Science) (Boston, MA, USA) against the *S. fruticosa* proteome and Uniprot plant databases. Mascot was searched with a fragment ion mass tolerance of 0.060 Da and parent ion tolerance of 15 ppm. Scaffold software 2.1 (UC Davis, Davis, CA, USA) validated the MS/MS-based peptide and protein identifications based on an identity threshold of 95% or greater, which identified 834 total proteins. We have performed fold change tests to analyze how many differentially expressed proteins are identified between treatments with a *p*-value of less than 0.05 after Hochberg–Benjamini Correction.

### 3.9. Statistical Analysis

All parameters were analyzed with at least three replicates and expressed as mean and standard error. One-way analysis of variance (ANOVA) was carried out to determine if salinity affected various parameters significantly. A Bonferroni post hoc test was performed (*p* < 0.05) to highlight differences, if any, among mean values.

## 4. Discussion

### 4.1. Growth Stimulation and Physio-Chemical Attributes under Moderate Salinity

Most growth parameters, except length and root dry mass, of *S. fruticosa* were stimulated in moderate (300 mmol L^−1^ of NaCl) salinity under hydroponic conditions. Khan et al. (2000) and Hameed et al. (2012) [49,51] also reported a stimulation of *S. fruticosa* growth in moderate salinity under soil culture. Similarly, stimulation of growth in low/moderate salinity occurred in many other species of the genus *Suaeda* such as *S. splendens* (200 mmol L^−1^ of NaCl) [76], *S. altissima* (250 mmol L^−1^ of NaCl) [77], *S. monoica* (500 mmol L^−1^ of NaCl) [78] and *S. salsa* (50 and 100 mmol L^−1^ of NaCl) [79]. These findings thus show that *Suaeda* species are obligate halophytes with an absolute requirement for a specific salinity level for optimal growth.

Plants of *S. fruticosa* dealt efficiently with osmotic constraint of moderate salinity (300 mmol L^−1^ of NaCl) by lowering succulence and accumulating Na^+^ as a ‘cheap osmoticum’ [16] to adjust leaf osmotic potential. Similar results were reported for many other succulent halophytes such as *Halosarcia pergranulata* [80] and *Salicornia dolichostachya* [81]. The major contribution of inorganic solutes (i.e., ~80% by Na^+^ and K^+^) in sap osmolality, as in this example, is a common characteristic of euhalophytes, which may be accounted for by low metabolic cost for osmotic adjustment [82]. The accumulation of many other organic solutes such as glycinebetain also contributes to tissue osmotic potential as reported in different members of amaranthaceae, halophytes including *S. fruticosa* [16,49]. High RWC of *S. fruticosa* indicates low water saturation deficit and high leaf turgor in 300 mmol L^−1^ of NaCl that may contribute towards optimum growth performance under moderately saline conditions.

In this study, a 3-fold higher Na^+^ content in leaves compared to roots of *S. fruticosa* (along with low H_2_O_2_ and unaltered MDA) in 300 mmol L^−1^ of NaCl was observed. Most euhalophytes (especially of the Amaranthaceae family) accumulate more Na^+^ in shoots as compared to roots [83,84]. The adequate compartmentalization of Na^+^ in the vacuole seems to be useful to lower sap osmotic potential and turgor maintenance that results in cell expansion, which is essential for growth. In addition to its osmotic role, Na^+^ may also contribute as a functional nutrient in *S. fruticosa* as previously reported in some members of Amaranthaceae such as red beet and sugar beet [85]. The optimal growth of *S. fruticosa* in 300 mmol L^−1^ of NaCl could also be linked with Na^+^ essentiality for C4 photosynthesis as reported for *Amaranthus tricolor*, *Atriplex tricolor* and *Kochia childsii* [86,87,88].

Our data show that *S. fruticosa* plants did not experience nutrient (K^+^, Ca^2+^ and Mg^2+^) deficiency in 300 mmol L^−1^ of NaCl. Test plants increased selective absorption of K^+^ over Na^+^ (possibly by HAK5 and AKT1 function; [89,90] to keep K^+^ levels adequate for optimal plant growth and physiology under moderate salinity. Similarly, adequate levels of Ca^2+^ and Mg^2+^ in *S. fruticosa* might possibly result from the high selectivity of Ca^2+^ via CNGC or GLR [91] and Mg^2+^ via AtMGT1, AtMGT7 and AtMGT9 [92] and high- and low-affinity transport systems [93] under moderate salinity. Ca^2+^ increased in both root and leaf tissues, which might be linked to its signaling roles and contribution in biomass development [94]. Among tissues, Mg^2+^ content was high in leaves and may be ascribed to its roles in photosynthesis under moderate salinity.

Often, a positive relationship between photosynthetic performance and plant growth, as found in this study, occurs. The net photosynthetic rate (*A*) of *S. fruticosa* plants peaked in 300 mmol L^−1^ of NaCl and is in agreement with findings on other obligate halophytes such as *Sesuvium portulacastrum* [34] and *Atriplex portulacoides* [95]. Under moderate salinity, higher stomatal conductance (*gs*) was observed in *S. fruticosa* plants that allowed better carbon assimilation (as well as in-take) that coincided with higher net transpiration (*E*). Likewise, *E* increased transiently under moderate salinity in *Atriplex portulacoides* [95]. Higher water transpiration (high *E*) may also contribute to cooling, and in this context, *S. fruticosa* had additional heat dissipation mechanisms, as evident from unchanged non-photochemical quenching. Activation of multiple biochemical processes for heat dissipation may have an energetic/metabolic cost. In this study, optimal growth of *S. fruticosa* with concurrent higher *A* under moderate salinity hints at efficient utilization of photosynthetic carbon gain into biomass production. The maximum photochemical efficiency of PSII (*Fv*/*Fm*) of *S. fruticosa* plants grown under moderate salinity was not compromised and remained comparable to the non-saline control, as reported in many other halophytes such as *Sarcocornia fruticosa* [96] and *Salvadora persica* [97]. However, a decline in apparent photochemical efficiency (*YII*) and photochemical quenching (*qP*) indicates some damage in photosystem II in this study. Increases in Na^+^ uptake reportedly interfere with functioning of light-harvesting complexes and pigment biosynthesis [98]. In this study, we also observed a decline in chlorophyll content of *S. fruticosa* leaves under moderate salinity, which also resulted in saturation of the photosynthetic system at lower light (see *Is*). The electron transport rate (*ETR*) generally remained unaffected, which indicates the resilience of photosynthetic photochemical machinery under moderate salinity in *S. fruticosa*. Since optimal *A* was achieved in moderate salinity, we hypothesize that the lower chlorophyll content and early light saturation of the photosynthetic system might be an adaptation to control *ETR* surge, thus preventing ROS overproduction.

Exposure of *S. fruticosa* plants to moderate salinity resulted in lower levels of H_2_O_2_ (a common ROS) and MDA (a common oxidative damage marker) compared to non-saline controls. However, an earlier study reported a slight increase in H_2_O_2_ and MDA levels of *S. fruticosa* plants in 300 mmol L^−1^ of NaCl under soil culture [51], while moderate salinity of 200 mmol L^−1^ of NaCl led to lower H_2_O_2_ and unaltered MDA levels in the leaves of *S. salsa* [30]. Both H_2_O_2_ and MDA contents of the model halophyte *Eutrema parvulum* (synonym, *Thellungiella parvula*) shoots did not increase under moderate salinity of 300 mmol L^−1^ of NaCl [99]. Hence, it appears that moderate salinity does not inflict oxidative damage in obligate halophytes. In this study, levels of SOD, CAT, GPX and GSH decreased, whereas APX, GR and AsA remained unaltered in leaves of *S. fruticosa* under moderate salinity, which might suggest that low production rather than quenching could be responsible for low H_2_O_2_ under moderate salinity. Consistent with our finding, activity of SOD decreased and that of GR remained unaltered in 300 mmol L^−1^ of NaCl in a succulent halophyte *Halopeplis perfoliata* [100]. Likewise, activities of SOD and peroxidase decreased, and GR activity was unchanged in *Salvadora persica* in 250 mmol L^−1^ of NaCl [97]. Uncompromised/optimal *A*, *gs* and *ETR* in *S. fruticosa* under moderate salinity also indicate that there was no over- reduction of photosynthetic machinery, which is a major contributor of ROS in green leaves [23,70].

### 4.2. Growth and Physio-Chemical Attributes under High Salinity

Plants of *S. fruticosa* could endure as high as 900 mmol L^−1^ of NaCl salinity, which led to a reduction in shoot but not root growth under hydroponic culture. Similar results were reported for *S. fruticosa* plants in high salinity under soil culture [49,51]. Many other species of the genus *Suaeda* such as *S. monoica* (1350 mmol L^−1^ of NaCl; [78]), *S. altissima* (750 mmol L^−1^ of NaCl; [77]) and *S. salsa* (600 mmol L^−1^ of NaCl; [101]) could also endure high (≥600 mmol L^−1^ of NaCl) salinity with some signs of growth reduction. Hence, *Suaeda* species appear to be highly tolerant to salinity. Growth reduction of obligate halophytes under high salinity could result from high energetic demands for transport and sequestration of ions and synthesis of compatible solutes for osmotic adjustment [16], which is also evident from high dark respiration (Rd) in *S. fruticosa* under high salinity. Hence, growth reduction in obligate halophytes under high salinity could be an adaptation [16,23] rather than damage, as evident in most sensitive plants [13].

A decrease in leaf sap osmotic potential up to −5.2 MPa along with maintained tissue water content indicates adequate osmotic adjustment in *S. fruticosa* plants under high (900 mmol L^−1^ of NaCl) salinity. Such high tissue osmotic potential seems to be achieved by Na^+^ sequestration in vacuoles. In our data, we found contribution of inorganic solutes of about 97% (i.e., 90% and 7% for Na^+^ and K^+^, respectively). Thus, the energy requirement for the transport of Na^+^ between tissues (root to shoot) and within cells (cytoplasm to vacuoles) with concomitant accumulation of organic osmotica to achieve the required osmotic adjustment might be ascribed to reduced plant growth at 900 mmol L^−1^ of NaCl. Furthermore, despite increased Na^+^ in both plant tissues (indicating high Na^+^ influx in root and uptake towards shoots), limited plant growth in 900 mmol L^−1^ of NaCl does not appear to be caused by ion toxicity as oxidative damage markers did not increase. Increased root-to-shoot Na^+^ accumulation indicates that *S. fruticosa* plants restrict the Na^+^ entry in the xylem stream to protect photosynthetically active tissues, indicating generally unaffected plant photochemistry. Moreover, the contribution of reduced transpiration and stomatal conductance with lower Na^+^ uptake could not be neglected. Among selected nutrients, Mg^2+^ content declined in leaf tissue that may be linked to a decline in chlorophyll a content and photosynthesis rate. In this study, plants of *S. fruticosa* maintained K^+^ homeostasis probably by synchronized action of increased selective absorption and maintained selective transport of K^+^ over Na^+^. *Suaeda fruticosa* increased Ca^2+^ content especially in leaf tissue as reported in *Tecticornia* sp. [102]. High Ca^2+^ could be involved in Na^+^ transport via the SOS pathway [103] and inhibition of voltage independent cation channel [104,105]. In addition, increased Ca^2+^ could play a role in *S. fruticosa* plants for stress signaling [106], enzyme activation [107], and membrane and cell wall integrity [108] to survive in 900 mmol L^−1^ of NaCl.

In this study, the net photosynthesis rate (*A*) of *S. fruticosa* was reduced under high salinity. This decline in *A* could be linked to two possibilities. First, the decline in stomatal conductance (*gs*) and intercellular CO_2_ (*Ci*) may represent a stomatal limitation of photosynthesis. Secondly, the decreased chlorophyll and functions of PSII (qP and YII) may indicate biochemical limitations of photosynthesis. Decreased expression of some chlorophyll biosynthesis proteins such as geranylgeranyl diphosphate reductase under high salinity also support a decline in chlorophyll content. Whereas, unchanged or even enhanced expression of many proteins involved in CO_2_ fixation such as Ribulose bisphosphate carboxylase (RuBisCO) large-chain, RuBisCO large subunit-binding protein subunit beta and chloroplast carbonic anhydrase hint at adequate potential of CO_2_ fixation, thereby lower *Ci* under high salinity could be related to stomatal limitation. Thus, high salinity seems to impose a combination of stomatal and biochemical limitation in our test species, in agreement with findings on many other halophytes [33,109]. An unaffected *ETR* under high salinity reflects no sign of electron leakage for excitation of oxygen molecules to form ROS.

In this study, levels of SOD, APX, GPX and GSH increased, while CAT, GR and AsA levels remained unaffected in leaves of *S. fruticosa* under high salinity. Expression of Catalase-3 under high salinity was also comparable to the non-saline control. Similarly, activities of SOD and peroxidase were stimulated in *S. nudiflora* under 680 mmol L^−1^ of NaCl [110]. Activities of SOD and peroxidase increased, and CAT activity decreased in *S. crassa* under high (1000 and 1250 mmol L^−1^ of NaCl) salinity [111]. Likewise, exposure to 1000 mmol L^−1^ of NaCl caused an increase in activities of SOD, APX and GR but not of CAT in *Atriplex portulacoides* [95]. Hence, an increase in the SOD-based antioxidant system under high salinity is evident in many obligate halophytes. Unchanged *ETR*, *YNPQ* and *YNO* alongside a decline in photosynthetic CO_2_ assimilation (A) in *S. fruticosa* under high salinity also indicate a greater probability of superoxide radicle formation due to leakage of electrons from over-reduced photosynthetic machinery that thus warrants the importance of the SOD-based system under high salinity. Levels of MDA and H_2_O_2_ of *S. fruticosa* leaves did not increase under high salinity, which indicates absence of oxidative damage despite growth reduction in high salinity. Similarly, MDA levels in many other obligate halophytes like *Atriplex portulacoides* (800–100 mmol L^−1^ of NaCl; [95]) and *Salvadora persica* (750–1000 mmol L^−1^ of NaCl; [97]) also did not surge in high salinity. Hence, these data indicate that obligate halophytes including our test species minimize oxidative damage under high salinity by modulating antioxidants and photobiology.

## 5. Conclusions

Our results suggest that *S. fruticosa* is an obligate halophyte, as most biomass parameters improved under moderate (300 mmol L^−1^ of NaCl) salinity. High (900 mmol L^−1^ of NaCl) salinity caused a partial decline in some biomass parameters. Under moderate salinity, *S. fruticosa* plants showed effective osmotic adjustment with concomitant accumulation of Na^+^ in both roots and leaves. Accumulation of Na^+^ did not accompany nutrient deficiency, damage to photosynthetic machinery or oxidative damage in 300 mmol L^−1^ of NaCl. Under high salinity, plants showed a further decline in sap osmotic potential with higher Na^+^ accumulation, which did not concur with a decline in RWC, *Fv/Fm* and oxidative damage markers (H_2_O_2_ and MDA). In comparison to control plants, there were 22, 54 and 7 proteins in the optimal salinity treatment, and 29, 46 and 8 proteins in the high salinity treatment that were up-regulated, down-regulated and remained unchanged, respectively. Our findings suggest that higher energy cost, rather than ionic toxicity, may be the cause of the biomass decline observed in *S. fruticosa* at high salinity. Hence, these findings highlight important details about key physio-chemical adaptations of this succulent halophyte, which can be exploited in the future to enhance its productivity under high salinity conditions for sustainable agriculture and ecological restoration.

## Figures and Tables

**Figure 1 plants-13-01529-f001:**
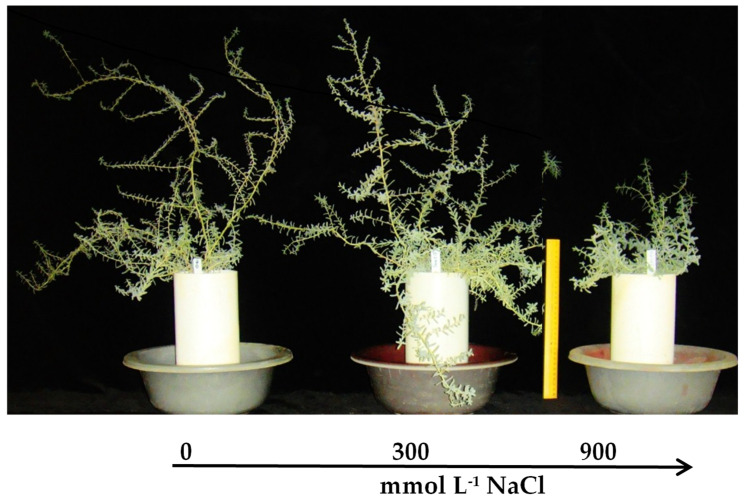
Plants of *S. fruticosa* grown under increasing salinity.

**Figure 2 plants-13-01529-f002:**
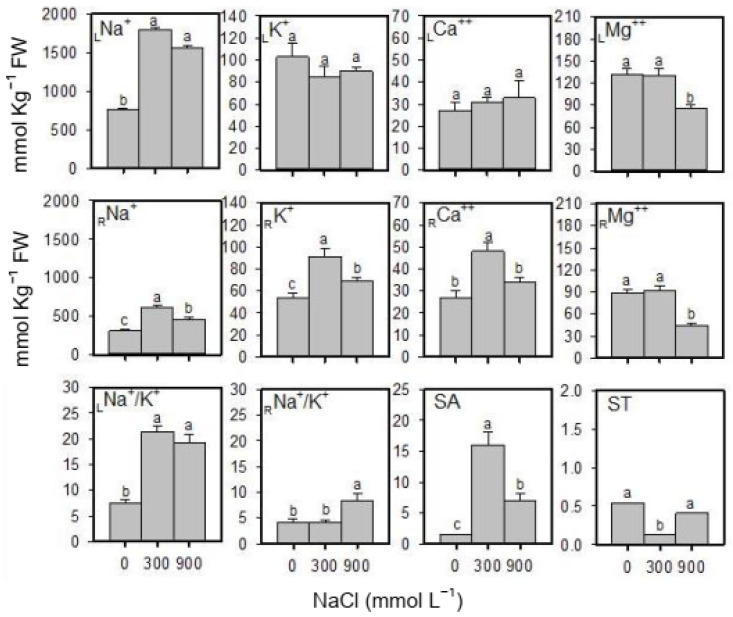
Various ion-relation parameters of the *S. fruticosa* plants under increasing salinity. Bars are mean ± standard error. Bars with different letters are significantly different from each other (Bonferroni test; *p* < 0.05).

**Figure 3 plants-13-01529-f003:**
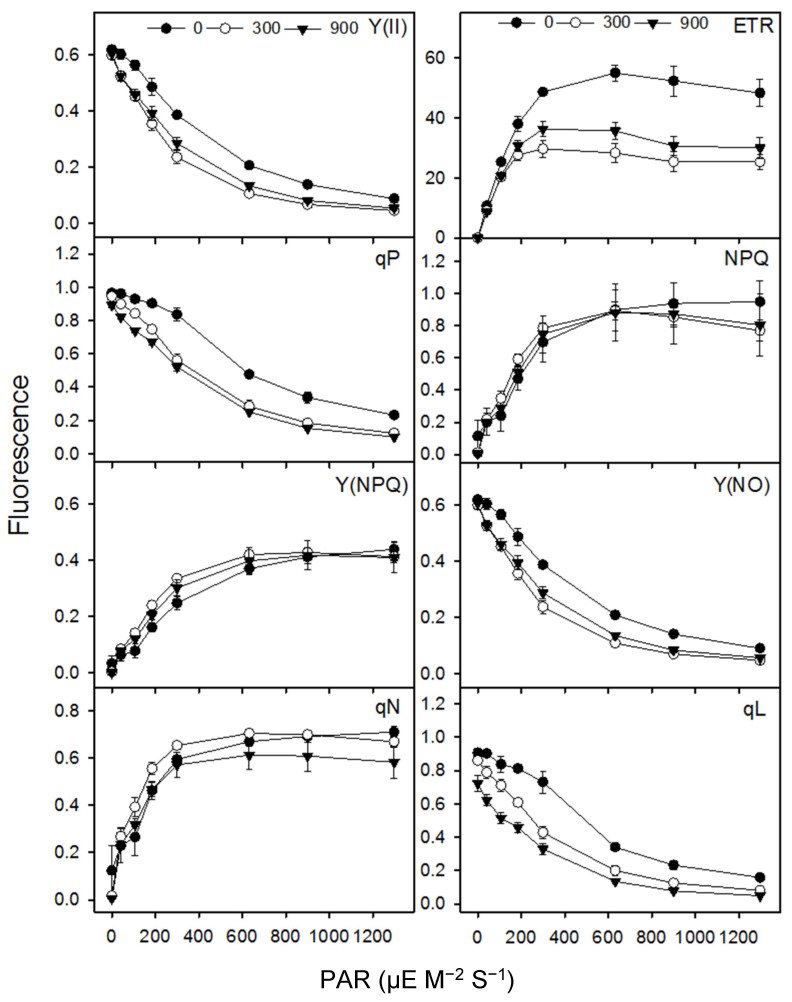
Chlorophyll fluorescence parameters of the *S. fruticosa* plants under increasing salinity. Values are mean ± standard error.

**Figure 4 plants-13-01529-f004:**
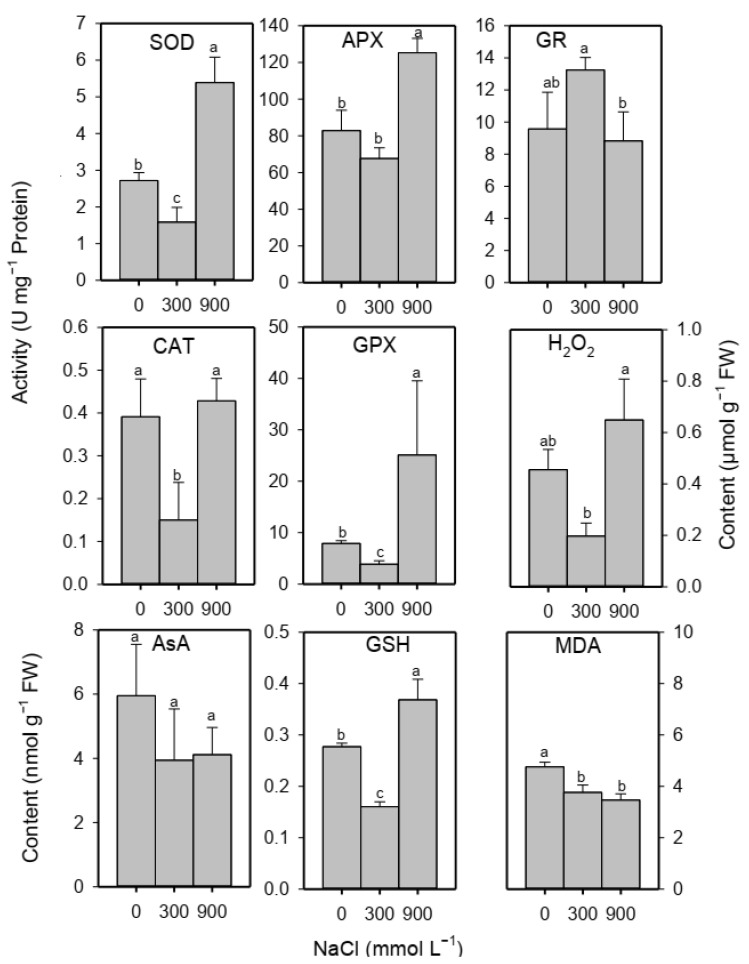
Levels of different antioxidant enzymes, substances and oxidative markers of the *S. fruticosa* plants under increasing salinity. Bars are mean ± standard error. Bars with different letters are significantly different from each other (Bonferroni test; *p* < 0.05).

**Figure 5 plants-13-01529-f005:**
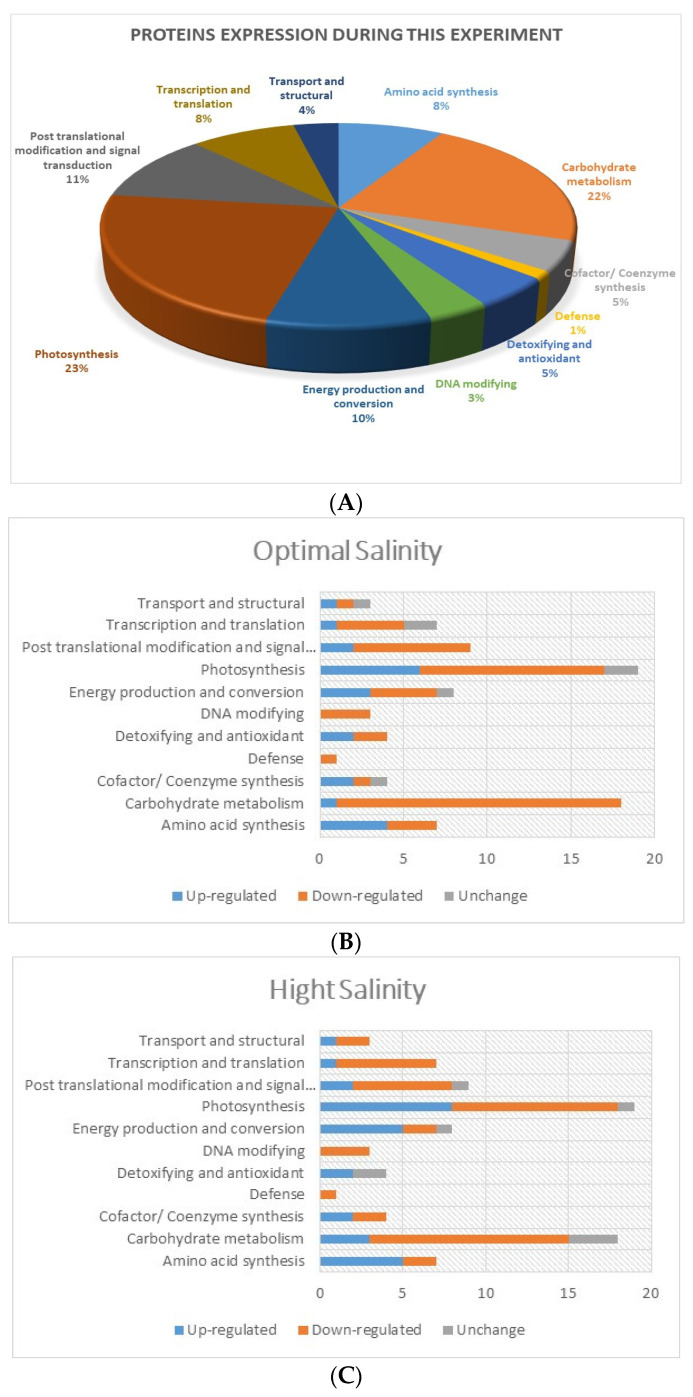
Proteomic responses of the *S. fruticosa* plants under different salinity. The overall protein expression in this experiment is presented in ten different functional categories of protein (**A**). The up-regulation, down-regulation and no change in these functional categories of proteins expression in optimal (300 mmol L^−1^ of NaCl) and high (900 mmol L^−1^ of NaCl) salinity are presented in (**B**,**C**), respectively.

**Table 1 plants-13-01529-t001:** Growth and water relation parameters of *S. fruticosa* plants under increasing salinity. Values are mean ± standard error. Values with different letters are significantly (*p* < 0.05; Bonferroni test) different across salinity.

	Salinity (NaCl; mmol L^−1^)		
	0	300	900
Plant Biomass			
lDM (g)	8.64 ± 1.0 b	12.63 ± 0.8 a	10.04 ± 0.9 ab
sDM (g)	10.10 ± 1.4 b	14.24 ± 0.8 a	4.02 ± 0.6 c
rDM (g)	2.16 ± 0.3 a	1.97 ± 0.2 a	2.47 ± 0.5 a
Plant Height			
rL(cm)	24.87 ± 2.5 a	27.37 ± 1.3 a	28.40 ± 0.9 a
sL (cm)	60.40 ± 1.8 a	58.63 ± 4.1 a	33.47 ± 5.0 b
Leaf Water Relations			
RWC (%)	86.31 ± 1.20 a	84.89 ± 1.30 a	90.28 ± 1.15 a
OP (MPa)	−2.11 ± 0.10 c	−3.38 ± 0.10 b	−5.19 ± 0.04 a
XPP (MPa)	−1.31 ± 0.07 c	−2.42 ± 0.08 b	−4.02 ± 0.02 a

**Table 2 plants-13-01529-t002:** Gas exchange, induction curve and photosynthetic pigments of *S. fruticosa* plants under increasing salinity. Values are mean ± standard error. Values with the same letter are not significantly different (Bonferroni test).

	Salinity (NaCl; mmol L^−1^)		
	0	300	900
Leaf Gas-exchange			
Rd	2.83 ± 0.1 b	2.25 ± 0.1 b	5.02 ± 0.2 a
Photo (A)	11.21 ± 0.1 a	13.71 ± 0.1 a	8.32 ± 0.3 b
Cond (gs)	0.03 ± 0.0 a	0.04 ± 0.0 a	0.03 ± 0.0 a
Ci	213 ± 3.8 a	213 ± 7.0 a	189 ± 2.6 b
E	1.72 ± 0.0 ab	2.12 ± 0.0 b	1.30 ± 0.0 a
WUE	6.52 ± 0.0 a	6.48 ± 0.1 a	6.41 ± 0.2 a
Ic	298 ± 2.4 b	211 ± 3.7 c	311 ± 1.4 a
Is	1960 ± 18.7 a	1576 ± 9.9 b	1455 ± 15.3 c
Leaf Photochemistry (Induction Curve)			
Fv/Fm	0.70 ± 0.03 a	0.73 ± 0.01 a	0.74 ± 0.00 a
Y(II)	0.11 ± 0.03 a	0.06 ± 0.01 b	0.08 ± 0.01 ab
Y(NPQ)	0.44 ± 0.04 a	0.45 ± 0.03 a	0.47 ± 0.03 a
Y(NO)	0.46 ± 0.05 a	0.49 ± 0.03 a	0.45 ± 0.02 a
NPQ	0.99 ± 0.17 a	0.93 ± 0.12 a	1.05 ± 0.13 a
qN	0.70 ± 0.03 a	0.67 ± 0.03 a	0.64 ± 0.03 a
qP	0.25 ± 0.07 a	0.14 ± 0.01 b	0.14 ± 0.01 b
qL	0.17 ± 0.05 a	0.08 ± 0.00 b	0.06 ± 0.01 b
ETR	26 ± 2.91 a	24 ± 1.76 a	30 ± 4.18 a
Photosynthetic Pigments			
Chl a	96 ± 7.1 a	52 ± 7.1 b	49 ± 10 b
Chl b	44 ± 10.0 a	19 ± 4.0 b	18 ± 5 b
Total	141 ± 17.1 a	72 ± 11.0 b	67 ± 14 b
CAR	25 ± 0.8 a	14 ± 2.1 b	12 ± 2 b
Chl a/b	2.29 ± 0.3 a	2.82 ± 0.3 a	3.00 ± 0 a
Chl/CAR	5.47 ± 0.5 a	5.12 ± 0.3 a	5.23 ± 0 a

## Data Availability

The original contributions presented in the study are included in the article/Appendix A, further inquiries can be directed to the corresponding author.

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
