# Peer review of "Thriving under Salinity: Growth, Ecophysiology and Proteomic Insights into the Tolerance Mechanisms of Obligate Halophyte *Suaeda fruticosa"

_plants, 2024, doi:10.3390/plants13111529_

Round 1

Reviewer 1 Report

Comments and Suggestions for Authors

Dear Editor,

The paper entitled "Thriving Under Salinity: Growth, Ecophysiology and Proteomic Insights into the Tolerance Mechanisms of Multipurpose Obligate Halophyte Suaeda fruticosa" fit very well into Special Issue intituled Wild Halophytes: Tools for Understanding Salt Tolerance Mechanisms of Plants and for Adapting Agriculture to Climate Change. Thus, the submitted manuscript represents a very valuable and intriguing work report focused on the proteomic analysis in addition to the biochemical and physiological techniques in order to better reveal the salinity tolerance mechanisms of the obligate halophyte specie Suaeda fruticosa. The topic is appropriate for the journal and the text is well written and inthis context, I believe that submitted manuscript is welcomed. Regarding the manuscript I reviewed,  I have only a few appreciations, observations and comments that might help readers understand the work better.

Abstract: concise, descriptive, emphasizing the idea, aim and results of the present study.

Introduction: very well written, well documented with relevant literature and it is able to introduce and familiarize with the subject of the study, in a very gradual and logical way.

Materials and methods: clear, logical and accurate. Perhaps is a little bit large, but I believe that this is the authors ‘choice to describe in very detail the used methods, comparatively to others, who only refer to the standard method, without giving many technical details. However, this is not in any case a weak point, but only a simple observation.

Observations:

1. Perhaps it would be good to titled paragraph 2.6. Oxidative damage markers instead of Oxidative stress markers. The same recommendation in subsection 3.5.

2. Pay attention to the numbering of the sub-chapters in the Material and methods because, inadvertently, there are two sub-chapters 2.6.

3. Pay attention to change the font of the text from Glutathione reductase from sub-chapter 2.6

Results: are presented in a concise, easy to follow manner, also using the tables and figures, which are well organized.   

Discussions enclosed precise comments based on obtained results.  Throughout this section, many references are being used, in order to support the given comments.

Author Response

Comment 1: “The paper entitled "“Thriving Under Salinity: Growth, Ecophysiology and Proteomic Insights into the Tolerance Mechanisms of Obligate Halophyte Suaeda fruticosa” only a few appreciations, observations and comments that might help readers understand the work better.”

Response: We thank the reviewer for the overall positive comments.

Comment 2: “Abstract: concise, descriptive, emphasizing the idea, aim and results of the …. without giving many technical details. However, this is not in any case a weak point, but only a simple observation.”

Response: Thanks.

Comment 3: “Observations: 1. Perhaps it would be good to titled paragraph 2.6. Oxidative damage markers instead of Oxidative stress markers. The same recommendation in subsection 3.5.”

Response: Changes made as advised (See titles of 2.6 and 3.5).

Comment 4: “2. Pay attention to the numbering of the sub-chapters in the Material and methods because, inadvertently, there are two sub-chapters 2.6.”

Response: Changes made accordingly. 

Comment 5: “3. Pay attention to change the font of the text from Glutathione reductase from sub-chapter 2.6”

Response: We have now changed the font of the highlighted text and ensured that the font is the same throughout the manuscript.   

Comment 6: “Results are presented in a concise, easy-to-follow manner, also using the tables and figures, which are well organized. ”

Response: Thanks.

Comment 7: “Discussions enclosed precise comments based on obtained results.  Throughout this section, many references are being used in order to support the given comments.”

Response: Thanks.

Reviewer 2 Report

Comments and Suggestions for Authors

Although there are previous studies on the salinity tolerance of Suaeda fruticosa, the species  belonging to a well-known genus of extremophilic plants, the manuscript is worthy of publication. This is an interesting multidisciplinary approach, combining physiological analysis with ion quantification, analysis of chemical antioxidants and oxidative stress markers, antioxidant enzyme activity completed by an extensive proteomic study, This approach is novel, the methodology and the result sections  are clearly explained. The discussion is appropriate, including adequate references. As in previous studies, the species is shown to be highly halo-tolerant but is affected by high salt concentrations of 900 mM NaCl. The novelty of the study is that it shows that the loss of biomass detected at this salinity is due more to a higher energy cost than to ionic toxicity.

In addition to improving the contrast in Figure 5, it is necessary to expand the legend and specify what each graph presented in this combined figure represents. 

Author Response

Comment 1: “Although there are previous studies on the salinity …….. of the study, it shows that the loss of biomass detected at this salinity is due more to a higher energy cost than ionic toxicity.”

Response: We sincerely appreciate the positive comments made by the reviewer.

Comment 2: “Besides improving the contrast in Figure 5, it is necessary to expand the legend and specify what each graph presented in this combined figure represents.”

Response: We have now changed the quality of Figure 5 and also added details in its legend.

Reviewer 3 Report

Comments and Suggestions for Authors

This study comprehensively explores the salt tolerance mechanism of Suaeda fruticosa, providing valuable data and insights for biological research on salt-tolerant plants and potential applications in salt-alkaline vegetation restoration.

1. In the section of Materials and methods, gradients of 300, 600 and 900 mmol L-1 NaCl have been set up, but why have the data for 600 mmol L-1 not been seen in the following sections?

 2. The font inconsistency in the determination of glutathione reductase in the Materials and Methods section needs to be corrected.

3. In Figure 2, it is recommended to change "Na" to "Na+", similarly for K, Ca, etc.

4. It is suggested that Figure 5 be divided into three sub-figures labelled 'A', 'B' and 'C'.

5. It is not recommended to separately discuss moderate salinity and high salinity in the Discussion section. It is suggested to analyze them together to ensure the coherence between your results and discussion, and organically connect them.

6. The conclusion should emphasise more the unique aspects of the salt tolerance mechanism of Suaeda fruticosa, in particular its strategies for maintaining life activities in high salt environments, and the potential significance of these findings for salt-tolerant plant research and applications.

Author Response

Comment 1: “1. In the Materials and methods section, gradients of 300, 600 and 900 mmol L-1 NaCl have been set up, but why have the data for 600 mmol L-1 not been seen in the following sections?”

Response: The 600 mmol L-1 NaCl level was mistakenly mentioned in the methods section and has now been deleted.

Comment 2: “2. The font inconsistency in the determination of glutathione reductase in the Materials and Methods section needs to be corrected.”

Response: We have now removed the font type inconsistency, as highlighted by the reviewer. 

Comment 3: “In Figure 2, it is recommended to change "Na" to "Na+", similarly for K, Ca, etc.”

Response: Change made.

Comment 4: “It is suggested that Figure 5 be divided into three sub-figures labelled 'A', 'B' and 'C'.”

Response: Change made.

Comment 5: “It is not recommended to discuss moderate and high salinity separately in the Discussion section. It is suggested to analyze them together to ensure the coherence between your results and discussion, and organically connect them.”

Response: We appreciate the reviewer's perspective. However, in the case of S. fruticosa, we found very distinct responses under moderate (300 mM NaCl) and high (900 mM NaCl) salinity. Therefore, we have discussed our results in this context to better explain plant performance under those two salinity levels separately. Then, in the conclusions, we have summarized the findings from the two salinity levels comparatively together.

Comment 6: “The conclusion should emphasise more the unique aspects of the salt tolerance mechanism of Suaeda fruticosa, particularly its strategies for maintaining life activities in high salt environments, and the potential significance of these findings for salt-tolerant plant research and applications.”

Response: The conclusion section already highlights the peculiar responses of S. fruticosa under moderate and high salinity. However, we have included some more details in light of the comment.     

Reviewer 4 Report

Comments and Suggestions for Authors

-What is the purpose of saying multiple purposes in the title? The title should be revised.

-Detailed information about S.fruiticosa should be given, Voucher number and identification should be stated.

-Is salinity caused only by NaCl? How compatible is the application with field studies?

-The humidity content and temperature of the environment should be detailed. Because if the temperature is too high in salinity, the salinity rate in the plant will increase due to the increase in transpiration, which may cause stress in the plant. This situation should be detailed.

-

Comments on the Quality of English Language

improved

Author Response

Comment 1: “What is the purpose of saying multiple purposes in the title? The title should be revised.”

Response: We have now revised the title in light of the comment.

Comment 2: “-Detailed information about S.fruiticosa should be given, and Voucher number and identification should be stated.”

Response: As advised, we have added details about S. fruticosa and provided the herbarium specimen voucher number (See Section 2.1). 

Comment 3: “-Is salinity caused only by NaCl? How compatible is the application with field studies?”

Response: A number of studies have reported that Na+ and Cl dominate in saline soils (Ismail et al., 2014; Shelke et al., 2019) and are also the two most common (constituting up to 85%) ions in seawater (Angelis, 2005). In this context, most physiological studies on the effects of salinity on plants use NaCl as a salinity treatment. For the sake of comparison with published studies, we also used NaCl. Since Na+ and Cl− dominate saline soils and are major causes of toxicity in plants, experimental studies with NaCl, including ours, can be thus used to explain many responses of plants under field conditions to some extent, especially if soil Na+ and/or Cll contents of the experimental area are known. 

  • Shelke DB, Nikalje GC, Chambhare MR, Zaware BN, Penna S, Nikam TD. Biotech. 2019 Mar;9(3):91. doi: 10.1007/s13205-019-1599-6. Epub 2019 Feb 18. PMID: 30800602; PMCID: PMC6385073.
  • Ismail A, Takeda S, Nick P. J Exp Bot. 2014;65(12):2963–2979.
  • Angelis, M.D. (2005). Major Ions in Seawater. In Water Encyclopedia (eds J.H. Lehr and J. Keeley). https://doi.org/10.1002/047147844X.oc1707

Comment 4: “-The humidity content and temperature of the environment should be detailed. Because if the temperature is too high in salinity, the salinity rate in the plant will increase due to the increase in transpiration, which may cause stress in the plant. This situation should be detailed.”

Response: Information about temperature and humidity is added in light of the reviewer’s comment (See Section 2.1).  

Comment 5: “- minor English editing.”

Response: Prof. Dr. Brent Nielsen (BYU-Provo, USA), a native English speaker, has improved the language of the manuscript.  
